

# Unveiling the mental state: validating the uBioMacpa Pro stress measurement tool among Chinese college students

Mingzhu Pan[1,2], Xinxing Li[3], Liying Yao[2], Solomon Gbene Zaato[4], Yee Cheng Kueh[5] and Garry Kuan[1]

[1] Exercise and Sports Science Programme, School of Health Sciences, Universiti Sains Malaysia, Kelantan, Malaysia
[2] School of Physical Education, Shangrao Normal University, Shangrao, China
[3] Department of Physical Education, Seoul National University, Seoul, Republic of South Korea
[4] Akenten Appiah, Menka University of Skills Training and Entrepreneurial Development (AAMUSTED), Kumasi, Ghana
[5] Biostatistics and Research Methodology Unit, School of Medical Sciences, Universiti Sains Malaysia, Kubang Kerian, Malaysia

Corresponding author
Garry Kuan, garry@usm.my

## ABSTRACT

Mental stress is a significant contributor to various health issues, including cardiovascular disease, anxiety, and depression. Scholars have developed many tools and methods to evaluate psychological stress states. uBioMacpa Pro is one of the measuring meters that evaluates accumulated stress by measuring heart rate variability (HRV). This study uses reliability and validity tests to validate uBioMacpa Pro among Chinese college students. A total of 260 students (females = 146, males = 114) with a mean age of 21 years (SD = 1.48, 1.51) were volunteers and recruited in the reliability and validity tests, respectively. The heart rate variability parameters showed satisfactory test-retest and inter-rater reliability, with the most intraclass correlation coefficient (ICC) values exceeding the acceptable threshold of 0.75. Validity assessment was done by exploring concurrent validity that measured the psychological stress of college students by using uBioMacpa Pro and using the validated Chinese version of the Stress Scales for College Students (C-SSCS) as a reference. The result showed a significant correlation between the uBioMacpa Pro stress index and C-SSCS questionnaire scores ($r = 0.246$, $p < 0.001$). The overall finding of our study implies that the uBioMacpa Pro has good reliability and validity, and it can be used for monitoring and assessing Chinese college students' mental stress.

## INTRODUCTION

Globally, mental stress is recognized as a critical factor affecting individual mental health and social function. The accelerated pace of societal transformation and the increasingly complex environment have posed significant challenges, reshaped lifestyles and intensified mental health concerns (*Ding, Liu & Xu, 2021*; *Yew et al., 2022*). Mental stress, as a prevalent mental state, profoundly impacts emotional stability, cognitive processes, and behavioral patterns while contributing to the development of physiological disorders such

as cardiovascular diseases and immune dysregulation (*Cohen, Janicki-Deverts & Miller, 2007*; *Salleh, 2008*). Consequently, exploring methods for assessing and intervening in mental stress is of paramount importance for enhancing the public mental health.

Mental stress refers to a negative psychological condition arising from an individual's perceived imbalance between internal and external resources when confronted with environmental challenges (*Lazarus, 1984*). In recent years, the issue of mental stress has become particularly pronounced among university students, emerging as a primary factor affecting their mental health. A sizeable portion of them suffer from various degrees of psychological disorders and are troubled by anxiety, depression, and other mental problems (*Hamaideh et al., 2022*; *Kovess-Masfety et al., 2016*; *Liu, Ping & Gao, 2019*). According to a survey conducted by *Venable & Pietrucha (2022)*, a significant proportion of students reported experiencing mental health symptoms on most days over the past year, including stress (66%), anxiety (54%), and self-doubt (50%). Even as the National Alliance on Mental Illness (NAMI) illustrated, more than 45% of college students dropped out due to mental health-related reasons (*Baldwin, 2018*). Survey participants were asked whether they had gone through a mental health crisis while on campus, and unexpectedly, 73% of respondents experienced it. In this survey, students expressed feeling stressed or overwhelmed about the course load, anxiety, panic, and depression about school and life, as they felt difficulty adjusting to a new routine and environment, all triggering their mental health crisis. A significant proportion of several undergraduate students in China also reported suffering from various mental health issues, including depression, obsessive-compulsive disorder, anxiety, and interpersonal sensitivity undergraduate students in China also reported suffering from various mental health issues, including depression, obsessive-compulsive disorder, anxiety, and interpersonal sensitivity (*Huang et al., 2021*; *Shan et al., 2022*). Thus, college students' mental health problems are becoming more and more common.

Presently, there is mounting research that has focused on the mental stress of college students. Therefore, some researchers have developed various techniques to evaluate mental stress states. The most popular method for assessing mental stress levels is to adopt subjective methods. Most researchers employ self-assessment questionnaires such as the Depression Anxiety Stress Scales (DASS), the perceived stress scale to assess the respondents' mental states (*Andrews & Wilding, 2004*; *Monroe, 2008*). Numerous studies have utilized questionnaire scores or self-reported assessments to evaluate mental stress levels (*Gao, Ping & Liu, 2020*; *Salleh et al., 2021*). However, self-reported measures are inherently subjective and susceptible to various biases, including respondent bias, acquiescence bias, and social desirability bias. These biases can lead to distortions in the accuracy of participants' responses, thereby compromising the validity of the data (*De Leeuw, 2012*; *Rada & Domínguez-Álvarez, 2014*). For instance, some respondents may fill in the questionnaire randomly to save time or not read the items carefully, resulting in inaccurate measurement of psychological stress. Compared to subjective methods, objective measurement tools based on physiological signals provide a more precise assessment of the physiological responses associated with mental stress, effectively addressing the limitations of traditional approaches.

Some researchers or healthcare organizations have developed physiological measurement tools to measure mental stress (*Katmah et al., 2021*). uBioMacpa Pro was one such tool. A Korean medical company developed this device. It evaluates accumulated stress by measuring heart rate variability (HRV) from pulse wave analysis of capillaries. HRV refers to the subtle fluctuations in normal heartbeats over time, regulated by the autonomic nervous system (ANS). It reflects the antagonistic interactions between the sympathetic nervous system (SNS) and the parasympathetic nervous system (PNS) (*Wang, 2019*). The ANS, comprising the SNS and PNS, plays a critical role in maintaining the balance essential for both psychological and physiological health (*Saboo, Kacker & Sorout, 2024*). HRV encompasses time-domain, frequency-domain, and non-linear metrics (*Piskorski & Guzik, 2007*; *Ziegler et al., 1999*). Common time-domain metrics of HRV include the standard deviation of normal-to-normal R-R intervals (SDNN), the root mean square of successive differences between adjacent R-R intervals (RMSSD), and the percentage of consecutive R-R intervals differing by more than 50 ms (pNN50). SDNN reflects overall heart rate variability and represents the combined contribution of both branches of the autonomic nervous system to heart rate regulation. RMSSD and pNN50 are recognized as reliable indicators of vagal activity. Frequency-domain metrics of HRV include total power (TP), very-low-frequency (VLF), low-frequency (LF), high-frequency (HF), and the ratio of LF to HF (LF/HF) (*Ziegler et al., 1999*). The LF-HF ratio is a widely recognized measure for assessing autonomic nervous system (ANS) balance, as it quantifies the relative activity of the sympathetic and parasympathetic nervous systems. In recent years, HRV metrics have found extensive applications in psychological research, serving as objective indicators for evaluating emotional states, monitoring stress responses, and assessing mental health (*Amra et al., 2023*; *McCraty & Shaffer, 2015*; *Saboo, Kacker & Sorout, 2024*). HRV measurement is a noninvasive and reliable technique for evaluating stress-induced physiological responses, serving as both a quantitative and qualitative assessment tool (*Rodrigues et al., 2018*). Contemporary, there are many stress analysis devices, such as Body Checker and SA-3000P. Compared to these devices, uBioMacpa Pro is lightweight, portable, and easy to measure. Health issues are becoming a growing concern as individuals are becoming more health conscious. Thus, anybody can quickly test accumulated stress status at home using the uBioMacpa Pro device to prevent further disease development.

Currently, some scholars are applying this device for related mental stress studies. *Lee et al. (2022)* used uBioMacpa Pro to investigate the effect of apartment community garden programs on stress. *Choi, Kim & Yun (2019)* also used this device to explore the impact of floral arrangement on the stress index of older people with chronic diseases. *Oh, Lee & Park (2021)* employed uBioMacpa Pro to determine the effect of stress level responses on indoor environmental color properties on heart rate variability. According to available research, Korea is where most current research on the usage of this gadget is concentrated. In China, there is a lack of sufficient research on the application of the uBioMacpa Pro. Given the cultural and environmental differences, it is crucial to validate this measurement tool within the Chinese population to ensure its applicability in measuring stress levels. Compared to complicated psychiatric examinations, which often require multiple sessions of clinical interviews, diagnostic assessments, and extensive

analysis, the uBioMacpa Pro offers a simpler, less time-consuming alternative for stress evaluation. Furthermore, psychometric questionnaires, such as the Daily Hassles Scale developed by *Lazarus (1984)*, contain 117 items, which require considerable time for participants to complete and for professionals to score and interpret, making them less practical for large-scale research. Thus, this study addresses this gap by validating the uBioMacpa Pro Stress Measurement Tool, which is essential for accurately assessing stress levels for Chinese college students, as well as providing a trusted measurement tool and reference for stress research in China.

## MATERIALS AND METHODS

### Ethical considerations

The study was approved by the Human Research Ethics Committee of Universiti Sains Malaysia (JEPem Code: USM/JEPeM/KK/23030207) and complied with the Declaration of Helsinki. Participants who met the inclusion criteria and voluntarily agreed to participate were provided with a research information sheet that detailed essential information, including the study objectives, procedures, potential risks, and benefits. Before the commencement of the study, all participants were required to complete the written consent forms to indicate their willingness to participate.

### Participants

This study involved university students in mainland China. Data were collected from 1st September 2023 to 1st October 2023 at Shangrao Normal University, Shangrao City, China. The promotional poster was distributed across the entire campus and shared on social media platforms, such as WeChat, to encourage voluntary participation. For those interested in the study, the researcher explained the study and sent an informed consent form containing detailed information. Participants would be screened for eligibility by researchers before joining the study. Eligible students were invited to enroll in this study and were required to fill out the informed consent forms. Finally, a total of 260 students volunteered to participate in the study. The study was divided into two parts: (i) A reliability test analysis and (ii) A validity test analysis of the uBioMacpa Pro. For the reliability test, 60 university students (30 males and 30 females) from Shangrao Normal University volunteered and participated in this study. For the validity test, the research team assessed the concurrent validity of uBioMacpa Pro. Accordingly, 200 university students (84 males and 116 females) were recruited for the validity assessment.

### Inclusion and exclusion criteria

The inclusion criteria for the study were that participants must be aged between 18 to 24 years, be willing to participate, and have no prior history of tobacco, alcohol, or drug use. Participants should also exhibit no skin breakdown, redness, swelling, or bruising on the index finger. The exclusion criteria included individuals with a history of chronic neurological or psychiatric disorders, those with no history of diabetes or hypertension, female participants who were menstruating at the time of the assessment, and individuals who felt unwell during the testing period. Participants were also informed that they could stop the study without any penalty.

## Procedures

This study employed a structured research design incorporating both quantitative and qualitative elements. Initially, the two uniformly trained researchers (A&B) tested the 60 participants using the uBioMacpa Pro device. Measurements were conducted daily from 8 a.m. to 5 p.m. in the kinesiology laboratory at Shangrao Normal University. Before measurements commenced, participants' basic information, including names, gender, age, academic year, and contact details, was collected. During the assessment, participants were seated comfortably for approximately 2.5 min to measure the stress index and heart rate variability parameters (low frequency (LF), high frequency (HF), autonomic balance (LF/HF), mean beat per minute (Mean BPM), standard deviation of NN intervals (SDNN), and root mean square of standard deviation (RMSSD)). After a 10-minute interval, participants retook the stress and HRV metrics assessment under the guidance of Researcher B to ensure inter-rater reliability. This measurement procedure was repeated over three consecutive days to evaluate test–retest reliability while participants remained blinded to their measurement results throughout the process. Subsequently, the research team employed uBioMacpa Pro alongside the validated Chinese Stress Scale for College Students (C-SSCS) to evaluate the stress levels of college students in assessing concurrent validity. The C-SSCS is a widely used tool for assessing subjective stress levels in Chinese university students. The objective stress index measured by uBioMacpa Pro was compared with the subjective stress scores obtained from the C-SSCS. Meanwhile, the researchers also recorded HRV parameters to investigate the association between stress levels and autonomic nervous system regulation. Both tools were administered to the same participants, and the data collected from these measurements were subsequently analyzed to determine the degree of consistency between the two instruments. This comparison aims to assess the alignment between the objective stress index provided by the uBioMacpa Pro and the subjective stress assessments derived from the C-SSCS, thereby confirming the concurrent validity of the uBioMacpa Pro in measuring stress levels. In addition to the content of C-SSCS, the questionnaire also collected participants' demographic information, including age, gender, and grade level. The flow diagram detailing the research procedure is illustrated in Fig. 1.

## Measures

All participants were administered a socio-demographic checklist (name, sex, age, major, health status, *etc.*) and were measured by objective and subjective mental stress measuring tools, uBioMacpa Pro and the C-SSCS, respectively. Based on the research design, the research team conducted a concurrent validity test of uBioMacpa Pro with C-SSCS.

### uBioMacpa device

The uBioMacpa Pro was developed by BioSense Creative Inc. in Seoul, Korea, in 2023 (Fig. 2). This innovative device measures heart rate variability (HRV) by analyzing pulse

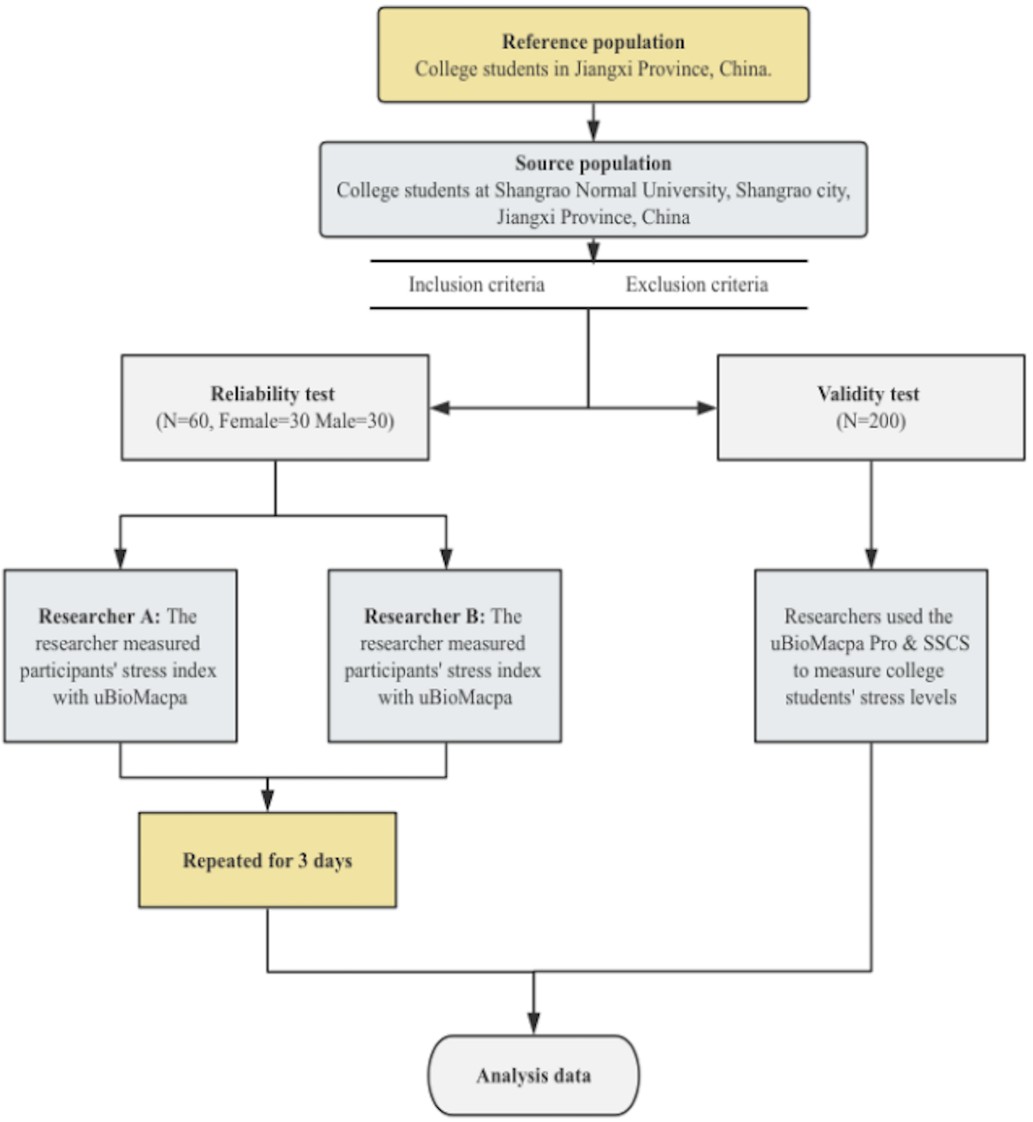

**Figure 1 The flow diagram outlining the research procedure.**

waves to assess accumulated stress. The stress test has two duration options: 2.5 min or 5 min. Typically, for testing accumulated stress, the duration is set at 2.5 min.

Participants were instructed to sit with their backs against the chair for 5 to 10 min to conduct the measurement. Before the measurement commenced, the researcher checked the participants' heart rates to ensure they returned to normal levels. During the measurement, the participant's index finger was clipped by the uBioMacpa Pro. The opposite end of the instrument was connected to the smartphone used for measurement through the data cable. Meanwhile, participants were instructed to remain still, avoid speaking, and avoid intentionally taking deep or shallow breaths.
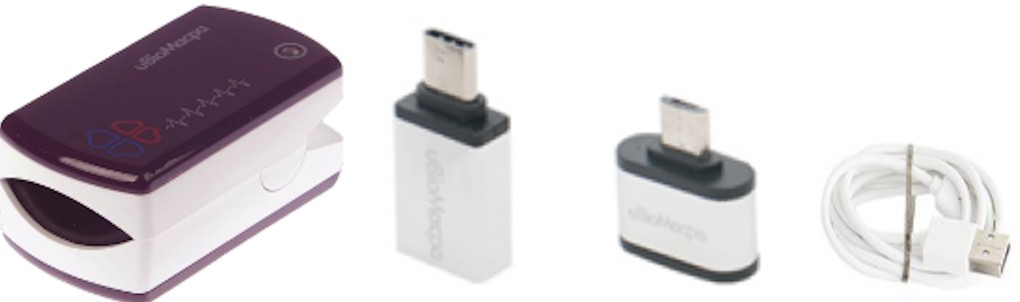

**Figure 2** UBioMacpa pro measuring probe.

Upon completion of the 2.5-minute measurement, the detailed report generated by the smartphone application provides the following data: pulse variability, along with a detailed analysis of testing parameters, including low frequency (LF), high frequency (HF), autonomic balance (LF/HF), mean beat per minute (Mean BPM), standard deviation of NN intervals (SDNN), and root mean square of standard deviation (RMSSD), as well as a participant-specific mental stress value, as shown in Fig. 3. The stress index was categorized into five groups to assess the varying levels of stress experienced by individuals. A score under 25 indicates 'nearly no stress', where individuals are generally emotionally stable and able to easily handle daily challenges. A score between 25 and 35 signifies temporary stress, typically triggered by short-term, situational stressors such as exams or deadlines, which lead to transient emotional responses but do not substantially impair functioning. A score ranging from 35 to 45 corresponds to primary stress, characterized by moderate, ongoing stress that may begin to affect cognitive and emotional well-being, though individuals remain capable of managing their responsibilities. Stress levels between 45 and 60 indicate accumulated stress with diminishing tolerance, where individuals experience more persistent, chronic stress that significantly strains their ability to cope, potentially resulting in fatigue, irritability, and reduced coping mechanisms. Lastly, a score over 60 represents chronic stress, reflecting a prolonged and severe state of stress that can have serious repercussions on both mental and physical health, often necessitating professional medical intervention to address its detrimental effects. This device had been tested by BioSense Creative Inc. on over 20,000 Korean participants and demonstrated good validity and high measurement accuracy.

### Chinese Versioned Stress Scale for College Students (C-SSCS)

The Chinese versioned Stress Scale for College Students (C-SSCS) consists of three subscales: Study troubles, Personal worries, and Adverse life events. Sample items include "Academic achievements are generally unsatisfactory," "Suboptimal physical appearance," " Poor family economic conditions," among others. The C-SSCS contains 30 items rated on a 4-point Likert-type scale (0 = no stress to 4 = severe stress), with total scores ranging from 0 to 90. A score of 45 or higher indicates high stress levels, while scores below 45 suggest low stress levels. The C-SSCS was developed by *Lee & Mei (2002)* and is a Chinese version

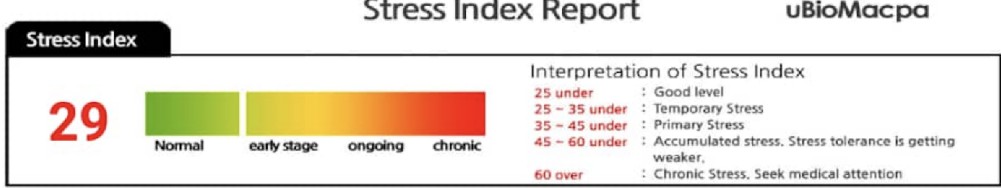

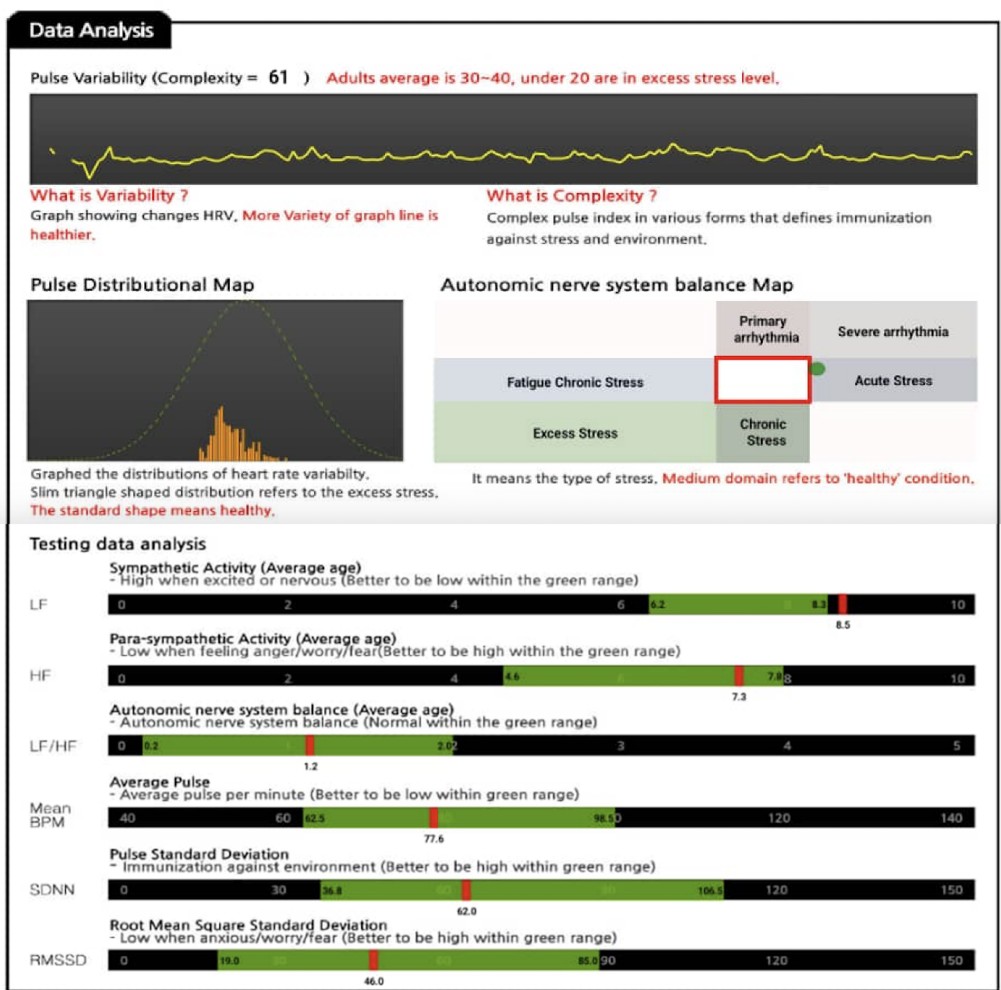

**Figure 3** Smartphone-generated stress index report for participants.

of the scale widely used in China. The scale has demonstrated good reliability and validity, with Cronbach's alpha values of 0.91 for the whole scale and 0.88, 0.84, and 0.83 for its three subscales. Additionally, the test–retest reliability coefficient for the overall scale was 0.78 (*Lee & Mei, 2002*).

## Statistical analysis

All of the measured data were entered into Microsoft Excel, mainly including the mental stress scores measured by uBioMacpa Pro and questionnaire data. This study was analyzed

**Table 1  Participant demographic characteristics.**

| Variables | M ± SD | N (%) |
|---|---|---|
| Age | 20.91 ± 1.48 | 60 (100%) |
| Male | 21 ± 1.44 | 30 (50%) |
| Female | 20.83 ± 1.51 | 30 (50%) |
| Freshman | – | 23 (38%) |
| Sophomore | – | 5 (9%) |
| Junior | – | 18 (30%) |
| Senior | – | 14 (23%) |

**Notes.**
M ± SD, Mean ± Standard deviation.

**Table 2  Reliability test results (ICC, 95%CI).**

| | Test–retest reliability | | Inter-rater reliability | | | |
|---|---|---|---|---|---|---|
| | A1A2A3 | B1B2B3 | A1B1 | A2B2 | A3B3 | AB |
| Stress index | 0.956 (0.93–0.97) | 0.962 (0.94–0.98) | 0.900 (0.84–0.94) | 0.890 (0.81–0.93) | 0.944 (0.9–0.97) | 0.950 (0.91-0.97) |
| LF | 0.857 (0.81–0.92) | 0.935 (0.9–0.96) | 0.876 (0.78–0.92) | 0.779 (0.63–0.87) | 0.759 (0.6–0.86) | 0.881 (0.76-0.93) |
| HF | 0.904 (0.85–0.94) | 0.972 (0.96–0.98) | 0.939 (0.9–0.96) | 0.867 (0.77–0.92) | 0.769 (0.61–0.86) | 0.918 (0.86-0.95) |
| LF/HF | 0.823 (0.73–0.89) | 0.836 (0.75–0.9) | 0.873 (0.74–0.92) | 0.723 (0.54–0.84) | 0.668 (0.44–0.8) | 0.808 (0.68-0.89) |
| Mean BPM | 0.942 (0.91–0.96) | 0.944 (0.91–0.97) | 0.957 (0.93–0.98) | 0.952 (0.91–0.97) | 0.982 (0.97–0.99) | 0.985 (0.97-0.99) |
| SDNN | 0.968 (0.95–0.98) | 0.982 (0.97–0.99) | 0.958 (0.93–0.97) | 0.933 (0.89–0.96) | 0.887 (0.81–0.93) | 0.947 (0.91-0.97) |
| RMSSD | 0.940 (0.91–0.96) | 0.970 (0.95–0.98) | 0.942 (0.75–0.96) | 0.868 (0.77–0.92) | 0.821 (0.7–0.89) | 0.914 (0.82-0.95) |

**Notes.**
A represents Researcher A, B represents Researcher B. A1 represents Researcher A's first Day of measurement, and B1 represents Researcher B's first Day of measurement. A2A3, B2B3, by that analogy.
LF, low frequency; HF, high frequency; LF/HF, autonomic balance; Mean BPM, mean beat per minute; SDNN, standard deviation of NN interval; RMSSD, root mean square standard deviation.

utilizing SPSS 27.0 statistical software. Standard descriptive statistics would be used for all variables. Normal distribution measurement data were expressed as M ± SD. $p < 0.05$ was considered statistically significant. The intraclass correlation coefficient (ICC) was used to assess retest and intra-rater reliability. Moreover, this study tested the correlation between uBioMacpa Pro and SSCS scores using Pearson correlation analysis to determine concurrent validity.

# RESULTS

## Reliability test result

A total of 60 college students (30 males and 30 females) from Shangrao Normal University were recruited for this part of the study. Table 1 presents the participants' demographic characteristics, including age, gender, and grade. The average age was 20.91 (SD 1.48) years. About 38% of the participants were first-year students. All descriptive data were presented by mean score and standard deviation. To examine retest and inter-rater reliability, the ICC was calculated for different combinations of 3 measuring results of the same participant, as shown in Table 2.

The results of 3 consecutive days of measurement of the same subject by researchers A and B are shown in the table as A1, A2, A3, B1, B2, and B3. A and B represent the average of three times the results for the same participant measured by researchers A and B, respectively. A1A2A3 was used to calculate the retest reliability ICC values by using the results of three measurements from Research A. A1B1 was used to calculate the inter-rater reliability ICC values using the first measuring results of Research A and B. Just as well AB represents the average of three times measuring results taken by two researchers on the same participant to determine inter-rater reliability.

As presented in Table 2, the reliability analyses indicated strong consistency across both test–retest and inter-rater evaluations. For test–retest reliability, all ICC values exceeded 0.80, with most metrics, including stress index, HF, Mean BPM, SDNN, and RMSSD, demonstrating excellent reliability (ICC>0.90). Although LF and LF/HF showed slightly lower ICCs in some comparisons, their values remained within acceptable thresholds. Regarding inter-rater reliability, ICCs across individual rater pairs ranged from 0.668 to 0.985, with most parameters exceeding 0.80. LH/HF exhibited comparatively lower inter-rater reliability among all parameters with ICCs ranging from 0.668 (A3B3) to 0.873 (A1B1), suggesting moderate to good consistency. Conversely, indicators such as Mean BPM, stress index, SDNN, and RMSSD showed consistently high agreement across raters. Notably, ICCs calculated from the averaged data across three sessions (AB) were uniformly high, ranging from 0.808 to 0.985, further confirming the robust reliability of the measurement indicators across raters and over time.

## Concurrent validity result

Two hundred volunteers (84 males and 116 females) aged 18 to 27 years (20.88 $\pm$ 1.51; means $\pm$ SD) were involved in this part of the study. There were more female students (116/200, 58%) than male students (84/200, 42%). Approximately 57% of the volunteers were junior and senior students. The mean uBioMacpa Pro stress index for the 200 participants was 31.31 (SD = 11.09), with scores falling between 25 and 35, indicating temporary stress. Meanwhile, the mean C-SSCS score for the 200 students was 36.79 (SD = 9.324). The HRV parameters of subjects were also summarized in Table 3. The mean LF value was 7.74 (SD = 0.65, range = 6.0–9.1), and the mean HF value was 6.8 (SD = 0.61, range = 5.2–8.3). The LF/HF ratio averaged 1.12 (SD = 0.09, range = 0.9–1.3). The mean BPM was 89.86 (SD = 8.98, range = 70.0–116.3). The mean SDNN was 45.38 ms (SD = 15.99, range = 20.2–83.8), and the mean RMSSD was 37.79 ms (SD = 14.84, range = 12.0–75.3).

Concurrent validity measures how a new test compares against a validated test. As described above, the C-SSCS has been validated. Therefore, our study utilized Pearson correlation analysis to evaluate the relationship between the uBioMacpa Pro stress index and C-SSCS scores. As presented in Table 4, the results revealed a significantly positive correlation between the uBioMacpa Pro stress index and C-SSCS scores ($r = 0.246$, $p < 0.001$), supporting the concurrent validity of the uBioMacpa Pro. Regarding HRV parameters, LF was negatively correlated with mental stress ($r = -0.167$, $p < 0.018$), while HF showed a stronger negative correlation ($r = -0.55$, $p < 0.001$). The LF/HF

**Table 3  Socio-demographic characteristics of the sample.**

| Variables | M± SD | N (%) | Range |
|---|---|---|---|
| Age | 20.88 ± 1.51 | – | 18–27 |
| Female | – | 116 (58%) | – |
| Male | – | 84 (42%) | – |
| Freshman | – | 42 (21%) | – |
| Sophomore | – | 44 (22%) | – |
| Junior | – | 79 (39.5%) | – |
| Senior | – | 35 (17.5%) | – |
| Mental stress (uBio) | 31.31 ± 11.09 | – | 19–64 |
| <25 | – | 23 (11.5%) | – |
| 25∼35 | – | 67 (33.5%) | – |
| 35∼45 | – | 72 (36%) | – |
| 45∼60 | – | 37 (18.5%) | – |
| >60 | – | 1 (0.5%) | – |
| LF | 7.74 ± 0.65 | – | 6–9.1 |
| HF | 6.8 ± 0.61 | – | 5.2–8.3 |
| LF/HF | 1.12 ± 0.09 | – | 0.9–1.3 |
| Mean BPM | 89.86 ± 8.98 | – | 70–116.3 |
| SDNN | 45.38 ± 15.99 | – | 20.2–83.8 |
| RMSSD | 37.49 ± 14.84 | – | 12–75.3 |
| Mental stress (C-SSCS) | 36.79 ± 9.324 | – | 10–52 |
| Low stress | – | 166 (83%) | – |
| High stress | – | 34 (17%) | – |

**Notes.**

LF, low frequency; HF, high frequency; LF/HF, low frequency/high-frequency ratio; Mean BPM, mean beat per minute; SDNN, standard deviation of NN interval; RMSSD, root mean square standard deviation; Mental stress (uBio), Mental stress index measured by uBioMacpa Pro; Mental stress (SSCS), Mental stress score measured by C-SSCS scale.

ratio was positively associated with mental stress ($r = 0.422$, $p < 0.001$), indicating a shift toward sympathetic dominance. Additionally, Mean BPM showed a significant positive correlation with mental stress ($r = 0.858$, $p < 0.001$), suggesting that higher stress levels were associated with increased heart rate. Furthermore, SDNN ($r = -0.755$, $p < 0.001$) and RMSSD ($r = -0.809$, $p < 0.001$) were both strongly negatively correlated with mental stress, suggesting that higher stress levels were associated with reduced HRV.

## DISCUSSION

To the authors' knowledge, this is the first study to examine the reliability and validity of the psychological stress-measuring meter (uBioMacpa Pro) for measuring mental stress among Chinese university students. In this present study, test–retest reliability and inter-rater reliability were utilized to evaluate whether the instrument demonstrates acceptable reliability. For the reliability analysis, the findings demonstrated high reliability for most parameters, with intraclass correlation coefficients (ICCs) consistently exceeding 0.80 in both test–retest and inter-rater analyses, suggesting that the measurement system

**Table 4** Association of mental stress (uBioMacpa Pro) with heart rate variability and C-SSCS.

| Parameters | r | p-value |
| --- | --- | --- |
| LF | −0.167 | <0.018 |
| HF | −0.550 | <0.001 |
| LF/HF | 0.422 | <0.001 |
| Mean BPM | 0.858 | <0.001 |
| SDNN | −0.755 | <0.001 |
| RMSSD | −0.809 | <0.001 |
| Mental stress (C-SSCS) | 0.246 | <0.001 |

provides stable outputs across time and raters. test–retest reliability analysis revealed that all parameters exhibited ICC values above 0.80, indicating high temporal stability over repeated measurements. In particular, indicators such as the stress index, HF, Mean BPM, SDNN, and RMSSD demonstrated excellent reliability (ICC > 0.90), suggesting these measures present good consistency when assessed at different time points under similar conditions. LF and LF/HF also showed acceptable reliability, with ICCs above the 0.80 threshold in most comparisons, although their values were relatively lower compared to the aforementioned indicators. This is consistent with prior studies suggesting that frequency domain indices, particularly the LF/HF ratio, are more susceptible to inter-individual variability and external disturbances, thus potentially reducing their temporal stability (*Burma et al., 2021*). Inter-rater reliability, on the other hand, revealed a wider range of ICCs (0.66–0.98), reflecting variability across different evaluator pairs. Most indicators showed high agreement between raters (ICC > 0.80), particularly for parameters such as stress index, Mean BPM, and SDNN. However, the LF/HF ratio displayed relatively lower inter-rater consistency, with ICCs ranging from 0.668 (A3B3) to 0.873 (A1B1). The variability of the LF/HF ratio across raters may be partly explained by its known sensitivity to individual physiological states and other influencing factors, as reported in previous studies (*Burma et al., 2021*; *Shaffer & Ginsberg, 2017*). However, the ICC values calculated from the average of three measurement sessions ranged from 0.808 to 0.985, demonstrating a substantial improvement in inter-rater reliability. This finding aligns with prior methodological research suggesting that repeated measurements and averaging can effectively reduce random errors and rater bias, thereby enhancing the stability and reliability of assessment outcomes (*Koo & Li, 2016*; *Weir, 2005*). Notably, despite limited prior experience, the two evaluators, both university students with only brief training on the uBioMacpa Pro device, achieved high measurement consistency. This further confirms the device's ease of use and reliability in various hands. Taken together, these findings suggest that the uBioMacpa Pro is not only a reliable tool for measuring mental stress but also user-friendly and portable, making it suitable for a wide range of settings and researchers, even those with minimal prior experience.

Subsequently, as for the validity study, we measured the psychological stress of college students by using uBioMacpa Pro and the C-SSCS as a reference. The percentage of participants with high levels of mental stress using the C-SSCS scale accounted for 17% of all participants, whereas the proportion of high levels of mental stress measured by

uBioMacpa Pro accounted for 19% of all students. The scale and the uBioMacpa Pro yielded similar results for measuring mental stress. To further verify the concurrent validity of the uBioMacpa Pro, Pearson correlation analysis was conducted to examine the relationship between its stress index and C-SSCS scores. The results demonstrated a significant positive correlation ($r = 0.246$, $p < 0.01$), suggesting that the uBioMacpa Pro is capable of reliably reflecting psychological stress levels in a manner consistent with the C-SSCS. This finding aligns with other studies that have validated similar technologies, such as physiological signal monitoring and smartphone-based applications, which have demonstrated significant correlations with self-reported stress levels (*Þórarinsdóttir et al., 2019*; *Zahari et al., 2021*). Additional support for validity was drawn from the correlations between uBioMacpa Pro stress index and HRV metrics.

Specifically, negative correlations with HF ($r = -0.55$), SDNN ($r = -0.755$), and RMSSD ($r = -0.809$) indicate that higher stress levels are associated with decreased parasympathetic activity and reduced HRV. These results are consistent with previous studies reporting reductions in HRV during stress-inducing tasks (*De Vries et al., 2021*; *Saboo, Kacker & Sorout, 2023*). Furthermore, *Kim et al. (2024)* demonstrated that mental stress levels in daily life are associated with specific alterations in HRV metrics. Specifically, lower HF values, indicative of reduced parasympathetic modulation, were found to be significantly associated with higher levels of perceived stress. The positive correlation between LF/HF ratio and stress index ($r = 0.422$, $p < 0.001$) supports a shift toward sympathetic dominance during elevated stress. This is consistent with HRV research suggesting that the LF/HF ratio increases under psychological stress, reflecting decreased parasympathetic modulation and elevated sympathetic arousal (*Kim et al., 2018*; *Shaffer & Ginsberg, 2017*; *Saboo, Kacker & Sorout, 2023*). Additionally, Mean BPM showed a significant positive correlation with mental stress ($r = 0.858$, $p < 0.001$), indicating that higher stress levels are associated with increased heart rate, which aligns with the physiological activation of the sympathetic nervous system during stress (*Kim et al., 2018*; *Thayer et al., 2012*). Although LF was also negatively correlated with stress ($r = -0.167$), the interpretation of LF is complex, as it reflects both sympathetic and parasympathetic influences. Thus, LF alone may have limited utility as a standalone physiological marker of psychological stress (*Billman, 2013*). Collectively, these findings support the construct validity of the uBioMacpa Pro. This evidence provides a foundation for the potential application of the uBioMacpa Pro in broader mental health monitoring and stress management strategies within university populations.

In scientific research, verifying the reliability and validity of measurement tools is a critical step in ensuring their quality and applicability (*Foxman, 2012*). From what has been discussed above, the results of the reliability and validity study of uBioMacpa Pro indicate that it has good reliability and validity among Chinese college students. These results suggest that the uBioMacpa Pro can serve as a reliable and practical instrument for assessing mental stress in this population. The implications of these findings are significant, particularly for the psychology management departments of Chinese universities. By utilizing the uBioMacpa Pro, institutions can gain a more accurate understanding of students' psychological stress levels, enabling the development of targeted

interventions and support mechanisms. Addressing psychological stress effectively requires consistent monitoring, making the availability of reliable measurement tools essential. Furthermore, it also offers Chinese researchers a reliable and standardized method for assessing psychophysiological conditions and stress indices across various populations. Its adherence to established validation guidelines and its ease of use make it a robust tool for studies focused on stress-related conditions, prevention strategies, and intervention effectiveness. These qualities might position the uBioMacpa Pro as a valuable tool not only for academic research but also for practical applications in psychological health management.

However, despite the immediate benefit of the instrument to mental health-relevant fields, the device's limitations should be noted. Several limitations exist in the measuring meter. For instance, uBioMacpa Pro occasionally produces a maximum or a minimum index during the measurement. For respondents with very low finger temperatures, the stress index may display as ''0'' in the report. Similarly, the device may not accurately record data for individuals with thin fingers or those wearing nail makeup, potentially limiting its applicability in certain populations. These challenges highlight the importance of controlling for environmental factors and individual conditions during data collection. Hence, researchers should ensure that measurements are conducted in a controlled environment and assess participants for exclusion criteria. By implementing these precautions, the reliability of the data collected using the uBioMacpa Pro can be further improved, enhancing its utility in both research and practical applications. In addition, one of the limitations of this study is the relatively small and homogenous sample, which impacts the broader applicability of the results. The sample was drawn exclusively from a single university, which introduces potential sampling bias and limits the generalizability of the findings. A larger and more diverse sample from multiple institutions or geographical regions would provide a more representative picture of the stress measurement tool's applicability and reliability across various populations. Furthermore, while this study offers valuable insights into the uBioMacpa Pro's effectiveness, it is important to recognize that the findings may not be fully applicable to broader, more diverse groups, and future studies should consider a more heterogeneous sample to enhance external validity.

## CONCLUSIONS

This article conducted an in-depth study on the reliability and validity of the mental stress measuring meter, uBioMacpa Pro, using Chinese university students as the study population. The findings confirm that the uBioMacpa Pro exhibits strong reliability and validity, establishing it as an effective tool for monitoring mental stress in this population. These results provide valuable insights for college mental health management, emphasizing the importance of accessible and standardized approaches to stress assessment. Future research should focus on validating the device across diverse populations and contexts to enhance its generalizability and applicability, as well as improving the precision of mental stress assessments to better support mental health monitoring and intervention strategies.

### Funding

This research was funded by the Educational Reform Project of Shangrao Normal University (Grant No. JG-23-18). The funders had no role in the study design, data collection and analysis, decision to publish, or preparation of the manuscript. The funders had no role in study design, data collection and analysis, decision to publish, or preparation of the manuscript.

### Grant Disclosures

The following grant information was disclosed by the authors:
Educational Reform Project of Shangrao Normal University: JG-23-18.

### Competing Interests

The authors declare there are no competing interests.

### Author Contributions

- Mingzhu Pan conceived and designed the experiments, performed the experiments, analyzed the data, prepared figures and/or tables, authored or reviewed drafts of the article, and approved the final draft.
- Xinxing Li analyzed the data, prepared figures and/or tables, authored or reviewed drafts of the article, and approved the final draft.
- Liying Yao conceived and designed the experiments, prepared figures and/or tables, authored or reviewed drafts of the article, and approved the final draft.
- Solomon Gbene Zaato performed the experiments, prepared figures and/or tables, authored or reviewed drafts of the article, and approved the final draft.
- Yee Cheng Kueh analyzed the data, prepared figures and/or tables, authored or reviewed drafts of the article, and approved the final draft.
- Garry Kuan conceived and designed the experiments, performed the experiments, prepared figures and/or tables, authored or reviewed drafts of the article, and approved the final draft.

### Human Ethics

The following information was supplied relating to ethical approvals (*i.e.*, approving body and any reference numbers):

All human study protocols were approved by the Universiti Sains Malaysia (USM) Human Research Ethics Committee (USM/JEPeM/KK/23030207) and followed the guidelines of the Declaration of Helsinki.

### Data Availability

The raw data is available in the Supplementary File.

## Supplemental Information

Supplemental information for this article can be found online at http://dx.doi.org/10.7717/peerj.19830#supplemental-information.

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
