# Peer review of "Unveiling the mental state: validating the uBioMacpa Pro stress measurement tool among Chinese college students"

_PeerJ, doi:10.7717/peerj.19830_

## Round 0.1 · original submission · Major Revisions

Please address all the comments of both reviewers. Note that both reviewers have provided PDF documents as well

·

Basic reporting

1. The manuscript's topic is fascinating, novel, and deeply meaningful. The author's exploration of this area is of great importance. However, more effort is needed to refine the language and sentence construction to convey this significance fully. Paraphrasing is necessary in many places to justify the topic's importance.
2. What are the validation criteria of uBioMacpa Pro
3. Introduction: The current introduction lacks the necessary impact and novelty. I recommend a complete rewrite. To enhance the significance of the research, consider starting with a comprehensive background (introduction to Mental stress), followed by an explanation of the definition and prevalence of stress in the present context. Highlight the gap in the literature and then connect it to your current measures and the study's conclusion.
4. Methodology: It takes a lot of work to develop and validate a stress score, so the researcher deserves praise. However, more specifics of the study methodology are required. How were students selected, contacted, screened, and given their consent? Where students were gathered, it was challenging and practically impossible to gather students two times, 8 am and 5 pm. The methodology should include the uBioMacpa Pro score details and categorization. Any researcher who has successfully enrolled and maintained participants in a study should share with the readers the strategies they employed, as it is not an easy task. Include information about sociodemographic factors in the methodology. Include more details and validate the questionnaire. How does the author ensure that the participant does not have a prior history of tobacco, alcohol, or drug use? Line no 124 should be paraphrased. This line is ethically not right.
5. One lacuna of this study was the small sample size. It should be included in the limitation section. The author should consist of education level (Semester) and socioeconomic status in sociodemographic variables. These two points are measure predictors of stress
6. The author should specify which domain of heart rate variability was analyzed
7. Add operational definitions of different categorization
8. Information on concurrent validity will be included in the methodology
9. How does the author ensure the coherence of uBioMacpa Pro
10. Results: The current presentation is not up to the mark. I suggest adding more details and providing a more precise account of the components of the uBioMacpa Pro score in the result section. This information is crucial for readers to understand the findings fully
11. Add the result of heart rate variability domain-wise and correlate it with stress score and HRV
12. The author did not add HRV findings that were very important for this study
13. The discussion is poorly written, with many grammatical errors, poorly structured statements, and no logical sequence of ideas. It is presumed that the author(s) in the discussion wanted to provide the data from the previous research. If true, please discuss your(authors) results clearly and link to the earlier findings. Many lines are paraphrased and compared in the discussion with relevant studies with proper citation. The author rewrites the discussion and compares it with studies based on validating stress scores with relevant studies; the information presented in the discussion section doesn’t link.
14. The author should include studies of HRV and stress scores and compare them with present study findings

15. Conclusion—This could be a little longer. Two to three more sentences should better describe the study and its results.

16. The author should check all references according to journal guidelines

Experimental design

no comment

Validity of the findings

no comments

Reviewer 2 ·

Basic reporting

There are lots of places in the manuscript need to clarify. I suggest authors invite a native English speaker to proofread their manuscript. I won't list all the areas that need to be rewritten.

Abstract: please rewrite the participants' numbers. Do not separate all 260 participants into two parts. I will confuse readers.

Background:
Line 43. ...students' psychology... what is that meaning?
Line 47, ..."College Student Mental Health Report Best Colleges" What is this?
Line 51. "(Marjorie Baldwin, 2018). should not be presented like this.
Line 66, .... mental status? What is this?
Line 74, ... psychological stress... Sometimes you use mental stress, sometimes you use psychological stress. Please be consistent.
LInes 87-98, you justified why you did this study. But it would help if you justified why there needs to be more relevant research on uBioMacpa Pro in China. Specifically, why compared to complicated psychiatric examinations and time-consuming psychometric questionnaires, Where is the complicated part of the psychiatric examinations? Give some examples. And, why psychological questionnaires are time-consuming. You need to explain more.

Experimental design

Line 122, ....why "...those with no history of diabetes or hypertension..." are your exclusion criteria?
LIne 134, ...."...three consecutive days..." What paper or literature suggests this is an appropriate interval for test-retest? Why not two weeks? or one month?
Lines 148-160, I still don't understand how to use uBioMacpa to measure stress by your figures and description. Do you put uBioMacpa on participants' hearts, heads, or any part of the body? How to read the measurement values, how to read it, how to analyze it? Please describe it in detail.
Line 170, gives an item example for the "Chinese versioned Stress Scale for College Students (C-SSCS)."

Validity of the findings

Table 1 is not necessary. It is redundant from Table 3.
Table 2 needs notes to explain what the numbers in the table mean.
Table 4 is incorrectly presented.
Figure 1 The picture is not clear.

Additional comments

The discussion must be theoretical insightful and well-explained. What is your contribution to the current literature? What is your strength of the study?
Please address these points by subtitles.

Annotated reviews are not available for download in order to protect the identity of reviewers who chose to remain anonymous.

---

## Round 0.2 · Major Revisions

Please address the concerns of Reviewer 1. As you can see, they are expressing reservations as to how well you are engaging with the revision process. Please provide meaningful responses to the various concerns

·

Basic reporting

There are still many concerns about the presentation of the study methodology. It seems that the authors are just accepting reviewers' comments to get the manuscript published; I'm not sure whether the study was performed with that rigor. It is okay to have study limitations, and it is okay if the results are not as favorable as expected. Presenting clear and detailed explanations will more likely help to get it published than providing vague or too good-to-be-true descriptions.

Experimental design

details of health status
2. More details on heart rate variability (HRV) measurement are needed by analyzing pulse waves to monitor vascular health and assess accumulated stress. How does the author measure vascular health and assess accumulated stress levels
3. Which domain of heart rate variability was assessed author replied that it was not satisfactory
HRV
The author should read the mentioned lines
Heart rate variability (HRV) domains are the different ways to analyze HRV, including time-domain, frequency-domain, and non-linear analysis.
Frequency-domain HRV
• Total power (TP): The total variance in heart rate over the recording period
• Ultra low-frequency power band (ULF): Reflects circadian, neuroendocrine, and other rhythms
• Very low-frequency band (VLF): Reflects vasomotor changes and thermoregulatory influences
• Low-frequency band (LF): Reflects sympathetic and parasympathetic influences on heart rate
• High-frequency power (HF): Frequency activity in the 0.15–0.40 Hz range
• LF/HF ratio: The ratio of low-frequency to high frequency
Time-domain HRV
• Quantifies the amount of HRV during a monitoring period
• The simplest HRV calculation method
• Reflects the variability in the R-R interval over time
Non-linear HRV
• Quantifies the complexity and unpredictability of a series of IBIs
• Assesses the overall complexity and unpredictability inherent in HRV
4. How do these parameters assess pulse variation, heart rate distribution, and autonomic nervous system balance?
5. Autonomic nervous system balance is a broad term. The author will need to specify it.

Validity of the findings

6. In Table 3, the author should add details of the residential area, whether they are hostellers or day scholars. This point is an important predictor of stress levels
7. Revise these lines for conclusion, delete hypothetical facts: These results hold significant implications for university mental health management, offering an accessible and standardized approach to assessing stress

Additional comments

8. These points not satisfactory I first revision
1.Add the result of heart rate variability domain-wise and correlate it with stress
score and HRV.

2.The author did not add HRV findings that were very important for this study
3.The author rewrites the discussion and compares it with studies based on validating stress scores with relevant studies; the information presented in the discussion section doesn’t link.

4. The author should include studies of HRV and stress scores and compare them
with present study findings.

---

## Round 0.3 · Minor Revisions

Dear Authors:

Please attend this minor revision.

Regards

·

Basic reporting

These points are not addressed in this revision
Points in the first revision
1. Add the result of heart rate variability domain-wise and correlate it with stress
score and HRV.
2. The author did not add HRV findings that were very important for this study
3. The author should include studies of HRV and stress scores and compare them with present study findings.
According to the author's reply, I have chosen to focus specifically on stress values for analysis and did not utilize the detailed HRV data in this study. In the abstract, the author mentioned that uBioMacpa Pro is one of the measuring meters that evaluates accumulated stress by measuring heart rate variability (HRV). When accumulated stress is measured by HRV, why does the author not mention it? The significance of this study was increased when HRV values correlated with stress scores. It is very important to observe what is the relationship between two.

Experimental design

it is an interventional study

Validity of the findings

No issue in validation

---

## Round 0.4 · Minor Revisions

Reviewer 1 has identified a lack of data. Please note that it is PeerJ policy that all data needed to reproduce the analysis and conclusions must be provided.

·

Basic reporting

Author mentioned HRV variables, i.e., pulse variability, along with a detailed
analysis of testing parameters, including total power (TP), very low frequency (VLF), low
frequency (LF), high frequency (HF), autonomic balance (LF/HF), mean beat per minute (Mean
BPM), standard deviation of NN intervals (SDNN), and root mean square of standard deviation
(RMSSD), kindly provide complete data of HRV variables that will be required and correlation analysis that specifies stress score with HRV variables in a table for validation. Kindly add results in a table for reviewer understanding
Remaining comments addressed

Experimental design

Appropriate

Validity of the findings

It is requested the author kindly provide all data of HRV and specify domains and their variables
and data of correlation that specify stresss score with HRV variables in table for validation

Additional comments

Remaining points addressed

---

## Round 0.5 · accepted · Accept

I believe you have now addressed the remaining reviewer concerns re HRV data and am pleased to accept this manuscript for publication.

·

Basic reporting

Author address all comments kindly procced for further.

Experimental design

satisfactory

Validity of the findings

satisfactory

Additional comments

appreciation to the author for the hard work